# A Novel Auto-Synthesis Dataset Approach for Fitting Recognition Using Prior Series Data

**DOI:** 10.3390/s22124364

**Published:** 2022-06-09

**Authors:** Jie Zhang, Xinyan Qin, Jin Lei, Bo Jia, Bo Li, Zhaojun Li, Huidong Li, Yujie Zeng, Jie Song

**Affiliations:** College of Mechanical and Electrical Engineering, Shihezi University, Shihezi 832003, China; zhangjie77@stu.shzu.edu.cn (J.Z.); qinxy@shzu.edu.cn (X.Q.); 20192109044@stu.shzu.edu.cn (B.J.); 20202109065@stu.shzu.edu.cn (B.L.); 20202009021@stu.shzu.edu.cn (Z.L.); lihuidong@stu.shzu.edu.cn (H.L.); 20212009014@stu.shzu.edu.cn (Y.Z.); 20212009017@stu.shzu.edu.cn (J.S.)

**Keywords:** synthesis dataset, prior series data, fitting recognition, YOLOX, Blender, inspection robot

## Abstract

To address power transmission line (PTL) traversing complex environments leading to data collection being difficult and costly, we propose a novel auto-synthesis dataset approach for fitting recognition using prior series data. The approach mainly includes three steps: (1) formulates synthesis rules by the prior series data; (2) renders 2D images based on the synthesis rules utilizing advanced virtual 3D techniques; (3) generates the synthetic dataset with images and annotations obtained by processing images using the OpenCV. The trained model using the synthetic dataset was tested by the real dataset (including images and annotations) with a mean average precision (mAP) of 0.98, verifying the feasibility and effectiveness of the proposed approach. The recognition accuracy by the test is comparable with training by real samples and the cost is greatly reduced to generate synthetic datasets. The proposed approach improves the efficiency of establishing a dataset, providing a training data basis for deep learning (DL) of fitting recognition.

## 1. Introduction

A power transmission line (PTL) system is the infrastructure of a country, necessary for industrial development and people’s livelihood [1]. The key components of a PTL system consist of PTLs, fittings, towers, etc. Once one of these components of a PTL system fails, it would cause significant economic loss [2], therefore it is necessary to carry out planned inspections for early fault detection and maintenance [3]. At present, the inspection methods mainly include two kinds: human inspection and robot inspection [4].

With the development of inspection technology, people strive to develop advanced inspection robots to replace humans. There are mainly three types of advanced inspection robots: flying-mode robots (e.g., UAVs), walking-mode robots (e.g., dual-arm inspection robots), and hybrid-mode robots (e.g., flying–walking PTL inspection robots—FPTLIR) [5]. The flying-mode robots fly in the air and can cross obstacles with a flexible inspection view. The walking-mode robots walk along the PTLs with a stable inspection view. The hybrid-mode inspection robots can fly in the sky and walk along the PTLs, with the advantages of the previous two kinds of robots. However, the key technology of inspection robots to identify and position various fittings on the PTLs is the primary support for autonomous inspection, defect detection, and power maintenance [6,7]. Object detection methods based on deep learning (DL) can automatically extract features without processing complex images [8], showing a great potential in image recognition [9]. Moreover, the influences of light intensity, target scale, and environment on image recognition can be effectively reduced. DL has two research directions on image recognition: one-stage detection and two-stage detection. The representative methods of one-stage detection have YOLO series, SSD, etc. The representative methods of two-stage detection have R-CNN, Fast R-CNN, Faster R-CNN, etc.

Many research efforts to detect fittings using DL have been reported in recent decades [10,11]. The SSD-based mode could achieve automatic multi-level feature extraction to identify aerial images with complex backgrounds [12]. Exact region-based convolutional neural network (Exact R-CNN) and cascade mask extraction and exact region-based convolutional neural network (CME-CNN) were proposed in the literature [13]. These methods could solve the accuracies of existing detection methods limited by the complex background interference and small target detection. Rahman et al., [14] used YoloV4 to identify porcelain insulators on the towers and achieve good performance.

It is well known that training a DL model needs a large dataset. However, the volume of the dataset is often inadequate while training DL models for fitting recognition. The volume of a dataset is one of the key factors affecting the accuracy of a model [15]. These methods on the expanding dataset can be mainly divided into two categories. One is to expand using the real dataset. Song et al., [16] investigated the impacts of data augmentation methods on the accuracy of target recognition, in which the impacts of the data augmentation methods were significant such as Gaussian blur, scaling, and rotation. Chang et al., [17] focused on the training characteristics of the neural network using towers and aerial images as a background. Images of insulator segments were randomly stitched to simulate different lengths and types of insulators and RGB channels, lighting, and color were randomly varied to generate the dataset. Rahman et al., [14] combined various image processing techniques (e.g., data expansion, shimmer enhancement) to increase the volume of the training dataset and to improve the generalization capability of the insulator detection model. Another direction is to expand by building a synthesis dataset. Zidek et al., [18] generated a DL training dataset from virtual 3D models. The advantage of the approach was the swift preparation of a 2D sample training dataset from virtual 3D models to train models. The recognition accuracy was comparable with the trained model by real samples. Chen et al., [19] proposed a sample expansion method based on a combination of statute and 3D modeling technology. The high accuracy results were obtained when the ratio of real samples to simulated samples was 2.0. Madaan et al., [20] rendered synthetic wires using a ray tracing engine and overlayed them on 67 K images from flight videos to generate a synthesis dataset, with an average precision score of 0.73 on the USF dataset. Table 1 shows the key points of these methods.

Data acquisition is hard and costly for a complex environment around PTLs and it relies on some specialized equipment (e.g., UAVs, inspection robots). In addition, the weather is changeable. It is challenging to include all working conditions in the data (e.g., the image of a lake on a sunny evening with a backlight). To address these difficulties, we propose a novel auto-synthesis dataset approach for fitting recognition using prior series data. The synthesis rules are formulated based on prior series data (including type and position of fitting, viewpoint position, topography, season, and irradiance). The 2D images are rendered by the synthesis rules using advanced virtual 3D techniques. Finally, a synthetic dataset can be generated with images and annotations obtained by processing images using OpenCV (a standard image processing technique). The various attributes in the synthesis dataset (e.g., type and pixel coordinates of fitting, background environment) would be adjusted and expanded. The coverage of the synthesis dataset is more comprehensive than the real dataset. The proposed approach allows to extract the prior series data according to the working conditions and synthesizes the dataset with the characteristic properties of the working conditions.

The main contributions of this paper are as follows:Formulating the data synthesis rules based on prior series data. The prior series data are derived from complex working conditions; therefore, the synthesis rule possesses the characteristics of these working conditions.Integrating advanced virtual 3D technology to render 2D images. Synthetic images would not be subject to topographic restrictions such as real images, reducing the difficulty of acquiring image.Proposing a novel auto-synthesis dataset approach for fitting recognition using prior series data. The synthetic dataset with space–time properties generated by this approach is used to train the DL model, making the network easier to learn important features. The training results are biased toward identifying the fittings at the source of the prior information, improving the accuracy rate of target recognition.

This paper is organized as follows: Section 2 describes the prior series data and the approach for generating the synthetic dataset. Section 3 trains a DL model by the synthetic dataset and verifies the effectiveness and feasibility of this approach by tests. Section 4 provides the discussion. Conclusions are drawn in the last section.

## 2. Proposed Approach

Training a DL model to identify a fitting needs a large dataset. However, it is difficult to collect the fitting dataset on the PTLs, requiring manual upline or manipulating inspection robots. Some fittings for a PTL system are shown in Figure 1. Moreover, the cost of collecting the fitting dataset is more than a general dataset collection and manual annotation is time consuming and imprecise. To address these problems, we propose a novel auto-synthesis dataset approach based on the prior series data.

We take a section of a 35 kV PTL system in Xinjiang as our research object. The PTL system is distributed in Yanqi and Kuerle, passing through flatland, wetland, and gobi. The average ratio of cloudy–sunny–cloudy is 52%, 30%, 9% in these areas from 2011 to 2021, respectively. The percentage of severe weather (unsuitable for inspection) is small at 9%. Therefore, the impact of severe weather might be ignored. The details of the auto-synthesis dataset approach are shown in Figure 2. The dataset can be automatically generated based on the prior series data using Blender software and OpenCV library.

### 2.1. Prior Series Data

The synthesis rules are formulated from the prior series data, which reduce the generation of inspection-independent images in the dataset and assign space–time properties to the dataset. Therefore, the quality of the synthesized dataset can be enhanced and the amount of non-essential features are reduced.

When a robot inspects the PTLs, the motion trajectory of the robot is planned, and the collected images depend on this trajectory. Figure 3 illustrates the flying and walking trajectories of the FPTLIR; therefore, the prior series data would restrict the viewpoints of collected images on the trajectory. The images whose viewpoints are outside the trajectory can be covered to avoid the DL model learning the unrelated features.

In general, if there is a PTL system with *j* segments (a segment refers to two adjacent towers and the PTLs between them), the set of a prior series data can be expressed as
(1)$$S=\left\{S^{i}|i=1,2,3,\cdots ,j\right\},$$
where *S^i^* is the set of a prior series data for the *i*th segment, it can be obtained as
(2)$$S^{i}=\left\{P^{m}N|m=1,2,\cdots ,N\in N+\right\},$$
where *P^m^* denotes kinds of prior series data, *P^m^N* represents the *N*th prior series data in P*^m^*, and the set of prior series data of each segment can be described as
(3)$$S^{i}-P^{1}N_{1}/P^{2}N_{1}/P^{3}N_{1}/P^{4}N_{4}(N_{1},N_{2},N_{3},N_{4}\in N).$$

The structure of the prior series data is shown in Figure 4. Figure 5 illustrates the meanings of prior series data. Each segment of the PTL system includes four prior series data, namely: prior fitting series (*P*^1^), prior inspection-view series (*P*^2^), prior topography series (*P*^3^), and prior time series (*P*^4^). Prior fitting series (*P*^1^) includes: prior information, i.e., insulator type (*P*^1^*T*), insulator position (*P*^1^*P*), damper type (*P*^1^*TD*), damper position (*P*^1^*PD*), and other various fitting types and positions. Prior inspection-view series (*P*^2^) includes: viewpoint position (*P*^2^*V*). Prior topography series (*P*^3^) includes: the prior information, i.e., topographic type (*P*^3^*R*). Prior time series (*P*^4^) includes: season(*P*^4^*S*) and irradiance(*P*^4^*L*).

The prior fitting series includes the prior information of many types of fittings with a high similarity; thus, the insulator is used as an example in these fittings.

Some sets of prior series data are shown in Table 2. The four prior series data are independent in that all series data have derived prior information. In addition, they are also interrelated in that multiple series data can influence the synthetic dataset property. For example, the position relationship between the insulator and viewpoint is influenced by the prior fitting series(*P*^1^) and the prior inspection-view series(*P*^2^). The background of images is determined by the prior topography series(*P*^3^) and the prior time series(*P*^4^). Insulator type, insulator-viewpoint position relationship, and background are the three key parameters for constituting the synthetic dataset.

#### 2.1.1. Prior Fitting Series

The prior fitting series includes the insulator position *P*^1^*P* and the insulator type *P*^1^*T*. The insulator position includes the position coordinate and the direction of the insulator.

The voltage of the PTL system of our study is 35 kV. The PTL system is composed of several tension segments. Two tension towers are constituted into one tension segment and tangent towers are constructed between the tension segments. Furthermore, angle towers to substitute tension towers are used for PTL system steering with an angle of 0° to 90°. Angle towers and tension towers are equipped with suspension insulator strings and tension insulator strings, and the horizontal angle of the tension insulator string is *β* (0° < *β* < 90°). Each tangent tower also has suspension insulator strings with three or four pieces.

Insulators can be divided into disk insulators and bar composite insulators. The component of disk insulator includes a steel cap, an insulator, and a steel foot. The insulator of disk insulators is available in various materials: porcelain, tempered glass, and rubber. The bar composite insulators are composed of an umbrella sleeve, a core bar, and metal at both ends.

#### 2.1.2. Prior Inspection-View Series

The prior inspection-view series includes the viewpoint position *P*^2^*V* of the robot, which includes the position coordinates and the direction of the viewpoint.

The inspection-view is related to the inspection mode of the robot. The inspection robots include three modes: flying-mode, walking-mode, and hybrid-mode. The flying-mode robot performs the fixed-point air inspection paralleling the detection objects, and the walking-mode robot walks along the PTLs with a fixed inspection viewpoint. The hybrid-mode robot has the view of the walking-mode robot and the flying-mode robot. The view angle relationship is shown in the upper left of Figure 5.

#### 2.1.3. Prior Topography Series

The prior topography series includes the topographic type *P*^3^*R*.

The PTL passes through flatland, wetland, and gobi. The flatland is mainly planted with cotton, apricot, red date, wheat, corn, sugar beet, tomato, pepper, melon, and grape. The main plants of the wetlands are phytoplankton and aquatic vascular plants. The main topography of the gobi is saline and dunes.

#### 2.1.4. Prior Time Series

The prior time series includes the irradiance *P*^4^*L* and the four seasons *P*^4^*S*. This series data are mainly used to constrain the season and the irradiance of the prior topography series. Figure 6a shows the variation of average irradiance from sunrise to sunset on the first day of every month over the decade. The strongest irradiance is 1025 W/m^2^. The peak is usually at 2:00 pm each day in January, March, and June. Figure 6b depicts the daily distribution of the average strongest irradiance over the decade. The stronger irradiance mainly occurred around June.

### 2.2. Auto-Synthesis Dataset

An auto-synthesis dataset is generated, including the following steps:The 3D model of the insulator pieces with the material information is imported into Blender and the insulator pieces are assembled into insulator strings;A camera is created so that the coordinate system of the camera viewpoint coincides with the world coordinate system. The position and the angle of the insulator strings are corrected by the prior inspection-view series and the prior fitting series, and the view angle of the camera is also adjusted;Background-free images are outputted, then the high dynamic range images (HDRIs) are imported and adjusted to render images based on the prior time series and the prior topography series;The background-free image is processed using standard image processing techniques (Python OpenCV library) to generate object bounding boxes and annotate files with basic parameters (position/dimensions);A synthetic dataset is generated with images and annotations. The synthetic dataset is divided into a training set and a validation set.

Figure 7 depicts the flowchart of the auto-synthesis dataset approach.

The insulator piece and insulator strings are shown in Figure 8. According to the prior fitting series and national standard GB/T 7253-2005, the 3D model of the insulator piece is drawn and imported into Blender as the basic model. Insulators include porcelain insulators, tempered glass insulators, and composite insulators.

The position relationship between the FPTLIR’s viewpoint and the insulator is shown in Figure 9. The viewpoint coordinate system (o_r_-x_r_y_r_z_r_) coincides with the world coordinate system. The insulator piece is assembled based on the Blender scripting language and the insulator string is translated and rotated relative to the viewpoint. The movement of the insulator string is described by the rotation matrix given in Equation (4) and the translation matrix given in Equation (5). The camera position remains unchanged and only the view angle is changed.


(4)$$R_{y}=\left[\begin{array}{cccc} \cos\beta & 0 & \sin\beta & 0\\ 0 & 1 & 0 & 0\\ -\sin\beta & 0 & \cos\beta & 0\\ 0 & 0 & 0 & 1 \end{array}\right];R_{z}=\left[\begin{array}{cccc} \cos\gamma & -\sin\gamma & 0 & 0\\ \sin\gamma & \cos\gamma & 0 & 0\\ 0 & 0 & 1 & 0\\ 0 & 0 & 0 & 1 \end{array}\right],$$
where*R*_y_ *R*_z_ represents the four-dimensional rotation matrix of insulators,*β*, *γ* is the rotation angle of each additional generated view.
(5)$$T_{xyz}=\left[\begin{array}{cccc} 1 & 0 & 0 & t_{x}\\ 0 & 1 & 0 & t_{y}\\ 0 & 0 & 1 & t_{z}\\ 0 & 0 & 0 & 1 \end{array}\right],$$
where*T*_xyz_ represents the four-dimensional translation matrix of the insulators,*t*_x_, *t*_y_, *t*_z_ is the translation distance of each additional generated view.

For the walking mode of the FPTLIR, *β* is related to the insulator angle, 0 < *β* < 35°; *γ* is related to the angle of the PTL system steering, 0 < *γ* < 90°. The values of *t*_y_, *t*_z_ are shown in Table 3 and relate to the tower type. *t*_x_ > 0.3 m is related to the distance of the robot from the tower.

For the flying mode of the FPTLIR, the translation matrix can be added to the translation matrix of the walking mode, whose translation matrix is described by Equation (6)
(6)$$T_{F-xyz}=\left[\begin{array}{cccc} 1 & 0 & 0 & t_{x}+t\prime_{x}\\ 0 & 1 & 0 & t_{y}+t\prime_{y}\\ 0 & 0 & 1 & t_{z}+t\prime_{z}\\ 0 & 0 & 0 & 1 \end{array}\right],$$
where*T_F_*_-xyz_ represents the four-dimensional translation matrix of the insulator in the flying mode.*t*′_x_, *t*′_y_, *t*′_z_ is the translation distance of each additional generated view relative to the walking mode.


The view angle of the camera is adjusted to ensure that the insulator is within the output image. The view angles for rotation along the z_r_ and x_r_ axes are described by Equation (7). The synthetic dataset parameters are listed in Table 3.
(7)$$\{\begin{array}{l} \theta_{z}=\arccos\frac{t_{z}{}^{2}+t_{y}\sqrt{t_{x}{}^{2}+t_{y}{}^{2}}}{t_{x}{}^{2}+t_{y}{}^{2}+t_{z}{}^{2}}\\ \theta_{x}=\arctan\frac{\sqrt{t_{x}{}^{2}+t_{y}{}^{2}}}{-t_{z}}(t_{x}\geq 0)\\ \theta_{x}=-\arctan\frac{\sqrt{t_{x}{}^{2}+t_{y}{}^{2}}}{-t_{z}}(t_{x}<0) \end{array}.$$

Blender is an open-source modeling and rendering software providing an internal python language to control object angles, object positions, object materials, camera views, and HDRIs. The background-free 2D images are generated based on the Eevee engine. The environmental background of four seasons and irradiance are added in Blender based on the HDRIs file. The images are rendered based on the Cycles engine and all images are generated at 720 × 1280 resolution.

The bounding box (including width, height, and center coordinate of the bounding box) of the insulator is detected using the OpenCV based on the background-free images. It is written to the XML files that the object classification, the position and the size of the object bounding box, and the image height/width. Figure 10 shows the image generated by Blender and the process of generating the XML from the 2D image. The synthetic pseudo codes are listed in Algorithm 1.


**Algorithm 1.** Auto-synthesis dataset algorithm**def main():**         # Parameter initialization         **_init_()**         # Read hdri files         hdri_folder = **bpy.path.abspath**("//hdri")         hdri_file = [**os.path.join**(hdri_folder, f) **for** f **in os.listdir**(hdri_folder) **if f.endswith**(".exr")]         # Synthetic dataset**for** num_insulators **in range(**starnum_insulators, endnum_insulators):          # Assemble insulators according to the number of insulator strings                  **zoom**(num_insulators)                   **for** hdris **in** hdri_file:                           **for** k **in range**(1, num_hdri):                                     # Counting                                     num_image += 1                                     # Loading environment                                     node_environment, node_background, link1 = **hdri**(hdris)                                     # Adjustment of environmental parameters                                     **hdri_adjust**()                                     # Mobile insulator                                     **move**()                                     # Adjusting ambient light                                     **light**()                                     # Switching the CYCLES rendering engine                                     **bpy.context.scene.render.engine** = ‘CYCLES’                                     **bpy.context.scene.cycles.device** = ‘GPU’                                     **save**(Fi0 + str(num_image) + ".png")                                     # Output background-free image                                     **bpy.context.scene.render.engine** = ‘BLENDER_EEVEE’                                     **bpy.data.worlds["World"].node_tree.nodes["Background"].inputs**[1]**.default_value** = 1                                     **clear**(node_environment, node_background, link1)                                     **save**(Fi2 + str(num_image) + ".png")                                      # Clear nodes and cache images, unload hdri                                     **bpy.context.scene.world.node_tree.nodes.clear()****                                     img_remove()****                                     hdri_reload()****         ** #Automatic generation of annotation files          **cv_label()**

### 2.3. Dataset Evaluation

The selected HDRI scenes include flatland, wetland, gobi, and snow with an equalization ratio. The irradiance of the images is also equal.

The costs of a generated dataset mainly include the rendering cost and the annotation cost. Generating a synthesis dataset of 10,800 images takes 35 h, including 6.8 min of annotation. The costs of rendering images are counted in Figure 11. The synthetic images are rendered using the Cycles engine on the GTX 1660, rendering one image per 8.69 s on average. The background-free images are generated using the Eevee engine on the AMD R7 3700X CPUs, generating one background-free image per 3.07 s on average. There are 328,000 images of 91 classes of objects in the MS COCO dataset. However, the task of annotating this dataset is arduous. For example, it takes more than 20,000 h [25] to determine object classes presented in images of the MS COCO and the average is 16.4 s per annotation. However, the automatic annotation takes only 38 ms per piece, reducing time costs compared with the manually annotated dataset.

## 3. Experiments

### 3.1. Experiment Description

(1)Dataset: A synthetic dataset of 10,800 images with 720 × 1280 pixels is divided into a training set and a validation set to train a DL model. A dataset of 1200 real images with 2160 × 3840 pixels is used to test the trained DL model. The ratio of the training set, validation set, and test set is 8:1:1. The format of these datasets is COCO2017. Note that the format of our generated synthetic dataset is VOC format and needs to be converted for the COCO format. The object classification is a porcelain insulator.(2)Experimental configuration: The experiments are conducted based on the DL framework YOLOX. The computer configuration is AMD R7 3700X CPU, NVIDIA GTX-1660 with 8 GB of video memory, and 16 GB RAM. The operating system is ubuntu 18.04.(3)Defect detection criteria (evaluation criteria): Three widely used indexes are used to quantitatively assess the performance of defect detection methods: precision (*P*), recall (*R*), and achieved mean average precision (mAP). *P* is the percentage of true samples among all the samples that the system determines to be “true”. *R* is the percentage of “true” samples found among all true samples. *AP* represents the detection accuracy of a single category, and mAP is the average of AP for each category.

(8)$$P=\frac{T_{P}}{T_{P}+F_{P}},$$(9)$$R=\frac{T_{P}}{T_{P}+F_{N}},$$(10)$$AP=\int_{0}^{1}P(R)dR,$$
where *T*_P_ and *F*_P_ are the number of correctly and incorrectly positioned defects, respectively. *T*_P_ + *F*_P_ is the total number of located defects, and *T*_P_ + *F*_N_ is the total number of actual defects.

### 3.2. Training Results with YOLOX

YOLOX is a released model for target detection, exceeding the YOLO series in 2021. YOLOX has a fast detection speed and meets inspection robots’ performance needs.

The YOLOX network architecture has three components: backbone, neck, and head. The backbone based on the CSPDarknet53 is used for feature extraction. The neck is used for extracting some more complex features. The head is mainly used to predict the classification and location of targets (bounding boxes). Figure 12 illustrates the YOLOX network architecture. The structure is input–backbone–neck–head–output. The input is a 640 × 640 pixel image obtained by the warpAffine conversion of a 720 × 1280 pixel image. And the output is the object classification, classification probability, and localizations of bounding boxes.

The pre-trained model (YOLOX-s) is used to reduce the training time. Verification is performed every 1 epoch. All other parameters use the default official configuration. The model is trained with 300 epochs and the training time is 36 h 48 min. The mAP of the validation set is shown in Figure 13. The mAPval5095 on the validation set is 97.22%.

### 3.3. Comprehensive Performance Analysis

The trained model is tested on the test set with 1200 images and the images of the test set are taken from 60 video streams of the camera shots. The position relationship of the camera insulator is consistent with the position relationship of the viewpoint insulator. Few images in the test set are stitched by adding insulators from cropped shooting images to the aerial images. The graphical image annotation tool for annotating the test set is LabelImg. Some examples of the images with annotations are shown in Figure 14.

Experiments show that R, P, and F1 are 95%, 96.18%, and 0.96, respectively, and the mAPtest5095 reached 98.38% when the score threshold is 0.5. The evaluation process of the YOLOX-s detection model is shown in Figure 15. The results show that the trained model using a synthetic dataset can effectively recognize real samples.

## 4. Discussions

### 4.1. Synthetic Dataset vs. Real Dataset

It takes 11.6 s to synthesize an image and annotation for a dataset based on virtual synthesis techniques; this would take less time if better hardware were used. The average time to annotate an image is 38 ms in the synthetic dataset, which is much lower than the average annotation time of 16.4 s for the real dataset. Therefore, Synthetic datasets are easily obtained without considering the difficulty of the generation and annotation. However, it is costly and difficult to expand a fitting dataset of a PTLs. Once the volume of real data increases, the time and the economic cost also substantially increase relative to the synthetic dataset. In addition, dataset annotation can also output depth maps and instance segmentation labels with much better speed and accuracy than manual. When generating synthetic data, there is no human subjective error in annotation accuracy and the scale of synthetic data would be easily expanded with scalability and operability.

### 4.2. Potential for the Synthetic Dataset Based on the Prior Series Data

The synthesis of the dataset can contain any pose of the model and any environment, synthesizing some unimportant images if the dataset synthesis is not constrained. For fitting recognition, the synthetic dataset covering too wide a range is not conducive to DL training because the collect images with viewpoints outside the motion trajectory are unrelated features. For example, the walking-mode robots walk along a ground wire and the robots cannot collect images where the viewpoint is not on the trajectory. The viewpoints of images are not on their trajectory, causing the DL model to learn unrelated features, taking more datasets to train the network to improve accuracy. However, prior series data can achieve better training results with a smaller dataset. The disorderly and random datasets are endowed with space–time properties along the ground wire, showing the great scalability and tunability.

### 4.3. False Detections and Omissions

The trained model is more likely to detect the round-like white objects as porcelain insulators when the volume of the synthetic dataset is 6 K to train the DL model. When the volume of the synthetic dataset increases to 10.8 K, this false detection is significantly reduced and the missed detections also decrease. This false detection would also be easily solved during the deployment of the trained model. One reason is that the viewpoint of the test set images is not on the robot trajectory and the actual viewpoint is located in the sky. All objects are shrunk in the image except for the PTLs, reducing the number of round-like white objects. Second, the identifications of insulators in the video stream are continuous and false detections are often intermittent, which would be shielded from false detections by this feature. The trained model can more accurately detect the images of the background with vegetation and bare soil and almost no missed detection. Some images in the front view of insulators on the tarmac are missed detection because the series information of these images does not belong to the prior series data. However, there are few missed detections in the elevation view and top view directions of insulators on the tarmac, indicating good generalization of the model.

### 4.4. Comparison with Other Dataset Expanding Methods

A summary of the methods on the expanding dataset is shown in Table 4. The main comparisons with the related studies are as follows: (1) Our proposed approach of the synthetic dataset based on the prior series data could obtain a dataset with a large number of important features. The trained model using this synthetic dataset could identify targets in complex backgrounds, showing good generalization and accuracy. The synthesis dataset for these studies [18,19,20] contains unimportant features that may be learned by the DL model. (2) The expanding dataset based on real images [14,16,17,20] can significantly increase the volume of the dataset, but collecting real images is difficult, especially for the fitting dataset. Our dataset synthesis is completely virtual; therefore, the synthetic images would not be subject to topographic restrictions like real images, reducing the difficulty of generating images. (3) The trained model using the synthetic dataset for the study [18] has great accuracy for identifying regular objects in the single-color background. Our synthetic dataset based on prior series data could improve the recognition accuracy of the DL model in complex backgrounds since the synthetic dataset has a lot of important features.

## 5. Conclusions

In this study, a novel auto-synthesis dataset approach is proposed for fitting recognition using the prior series data. The generated synthetic dataset by this approach is validated on the YOLOX model. The main conclusions are as follows:The synthetic dataset is generated by the Blender script using the prior series data. Per 720 × 1280 pixel image and its annotation are generated in only 11.6 s. The efficiency of synthesizing the dataset is substantially improved compared with humans collecting images and annotating.The formulation of synthesis rules based on prior series data can control the properties of the dataset. The synthesis dataset has strong scalability, operability, and tunability.Training the YOLOX model using a synthesis dataset without real samples can obtain good models. The trained model achieves an mAP of 0.98 on the test set of real samples, indicating that the trained model on the synthetic dataset has a great generalization to recognize real samples. The research results suggest that training the DL model using a synthesis dataset is promising.

The limitations of the proposed approach mainly focus on two aspects. One is that the performance of our GPU device is not excellent and the time cost of generating a dataset will be very high if the volume of dataset needed is very large. We could replace the GPU with more excellent performance and optimize the algorithms in the future to reduce the time cost of the generated dataset. The other is that the depth maps and instance segmentation labels are not generated in this paper, and we will refine our program to generate them in the following work.

## Figures and Tables

**Figure 1 sensors-22-04364-f001:**
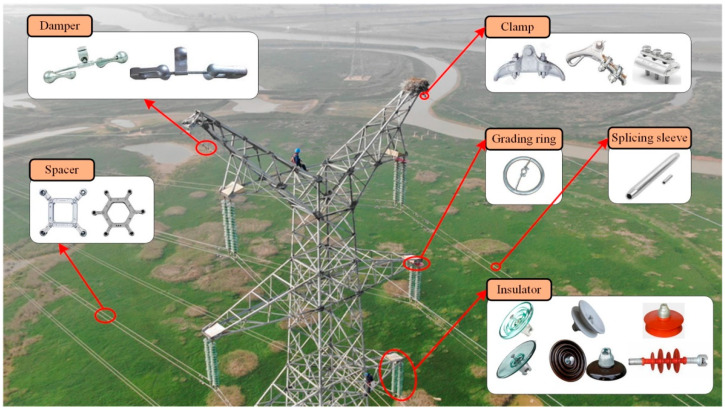
Illustration of fitting type.

**Figure 2 sensors-22-04364-f002:**
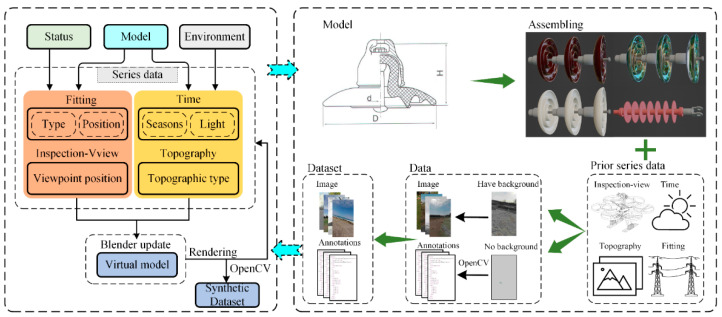
The details of the dataset auto-synthesis approach.

**Figure 3 sensors-22-04364-f003:**
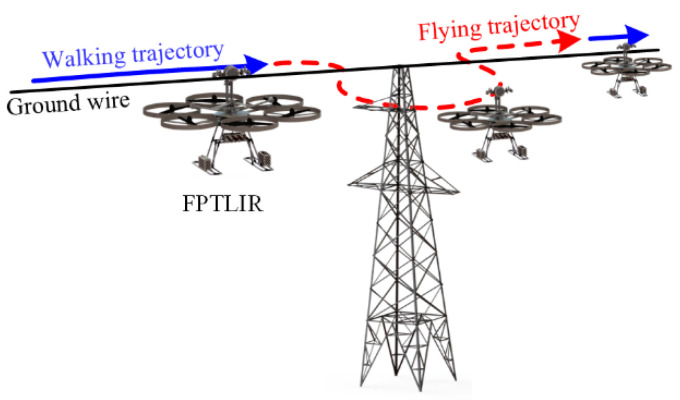
Illustration of the motion trajectory of the FPTLIR.

**Figure 4 sensors-22-04364-f004:**
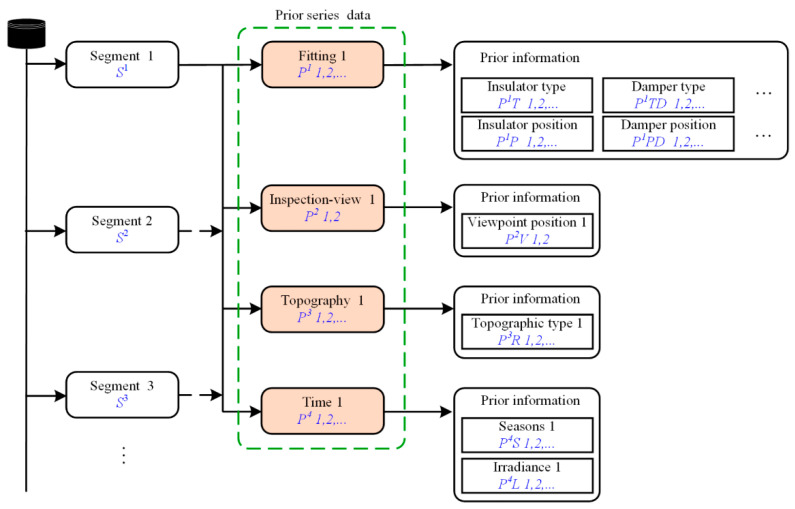
Illustration of the structure of the prior series data.

**Figure 5 sensors-22-04364-f005:**
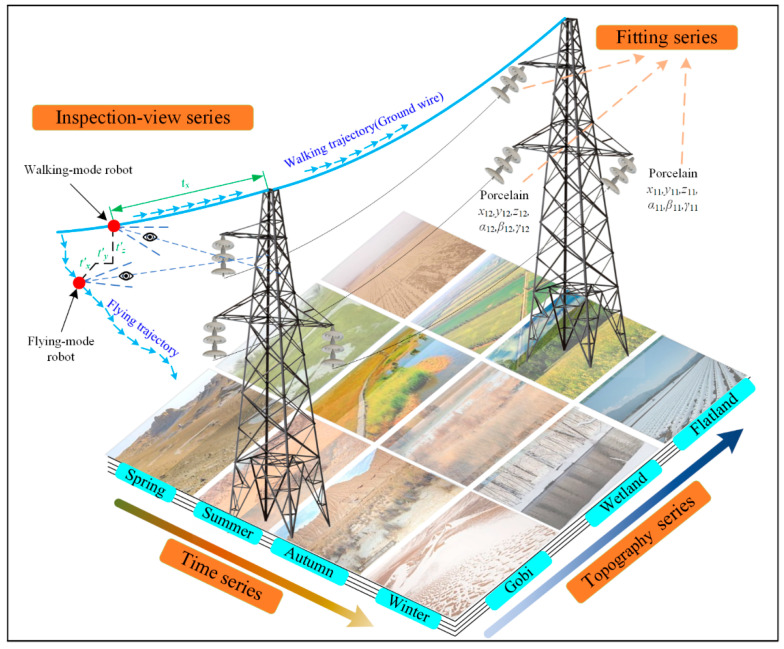
Illustration of the meanings of a prior series data in a segment.

**Figure 6 sensors-22-04364-f006:**
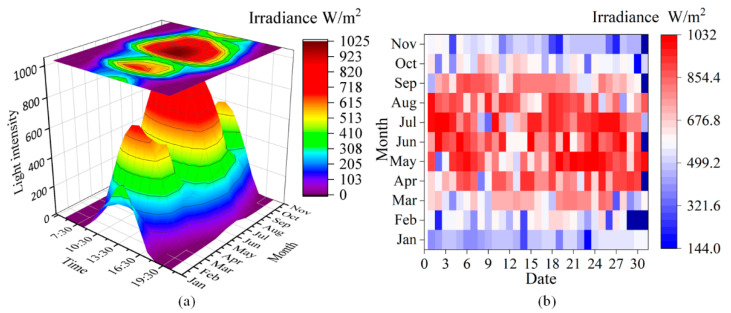
(**a**) Variation of day irradiance from light to no light on the first day of each month, (**b**) daily distribution of the strongest irradiance.

**Figure 7 sensors-22-04364-f007:**
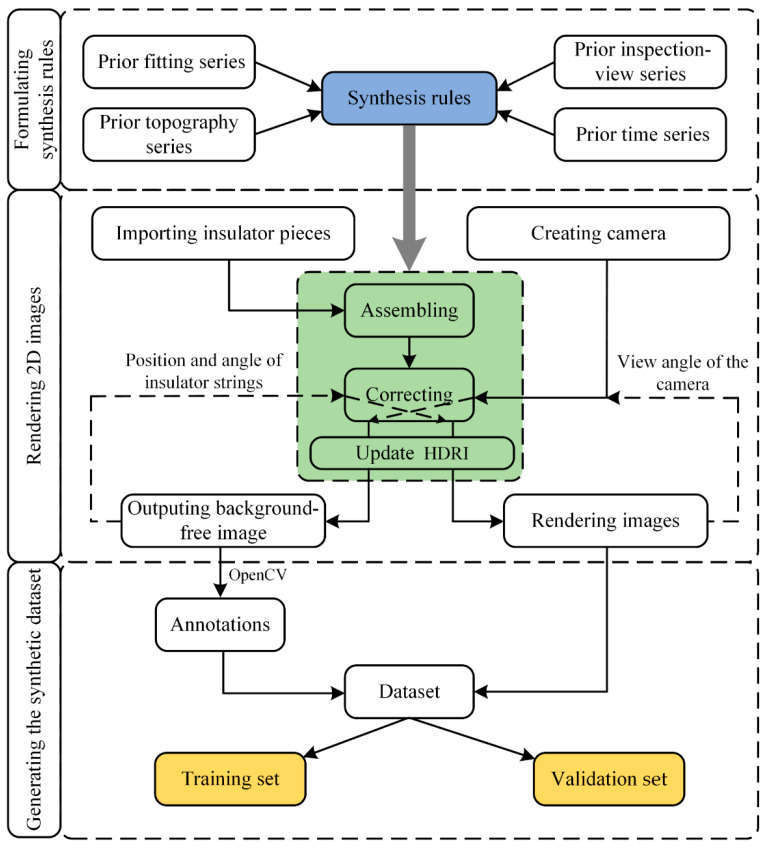
Flowchart of the auto-synthesis dataset approach.

**Figure 8 sensors-22-04364-f008:**
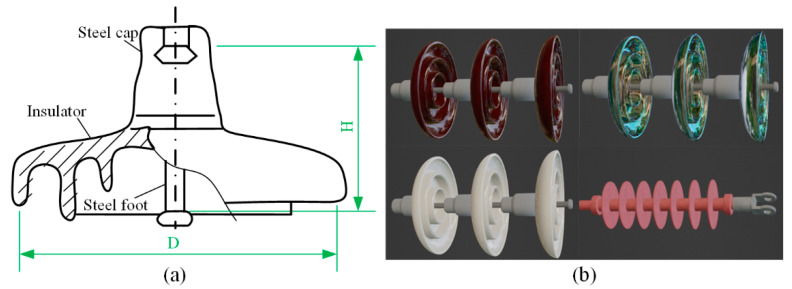
(**a**) Structure of insulator piece. (**b**) Four types of insulator strings in Blender.

**Figure 9 sensors-22-04364-f009:**
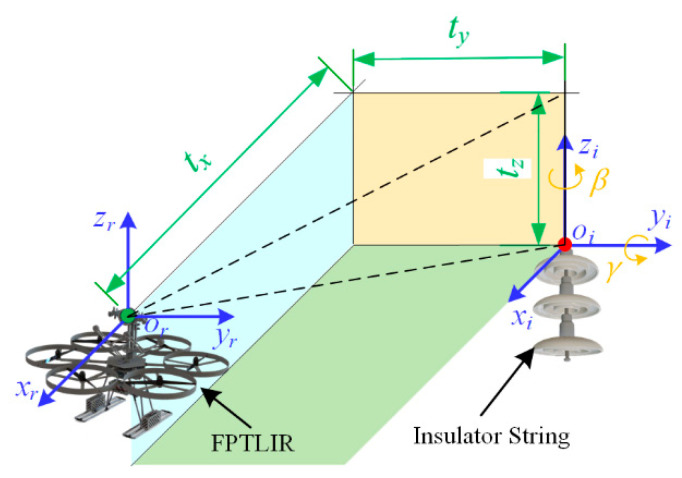
The position relationship between the FPTLIR’s viewpoint and the insulator.

**Figure 10 sensors-22-04364-f010:**
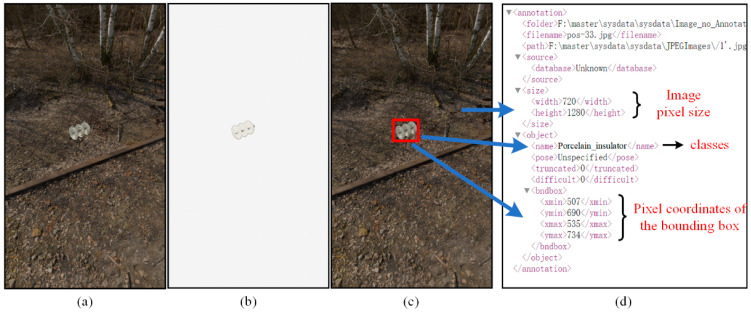
(**a**) Rendered image in Blender. (**b**) Generating backgrounded-free image in Blender. (**c**) Drawing bounding boxes on the rendered image, processing background-free image by OpenCV to get the bounding box. (**d**) Key image parameters in the XML file.

**Figure 11 sensors-22-04364-f011:**
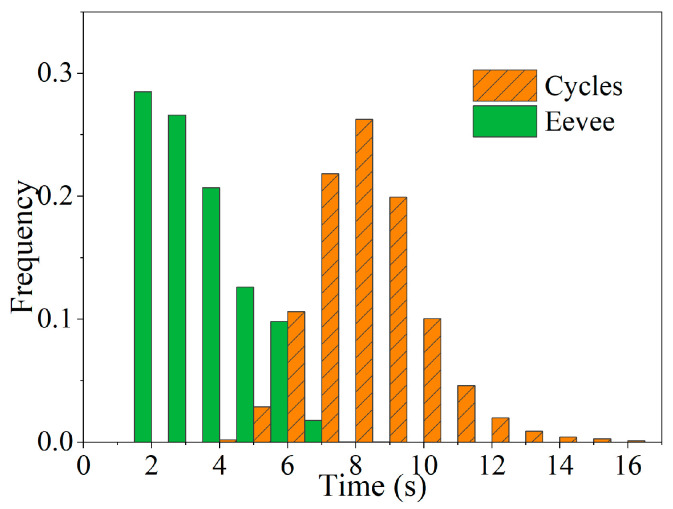
Costs of rendering images.

**Figure 12 sensors-22-04364-f012:**
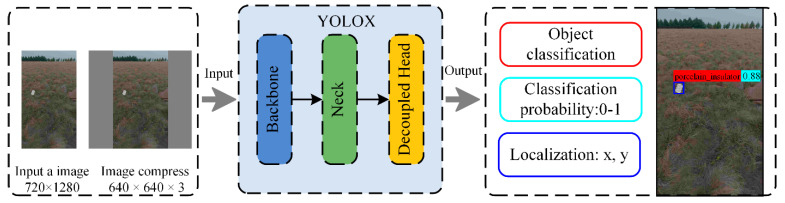
Illustration of the YOLOX network architecture.

**Figure 13 sensors-22-04364-f013:**
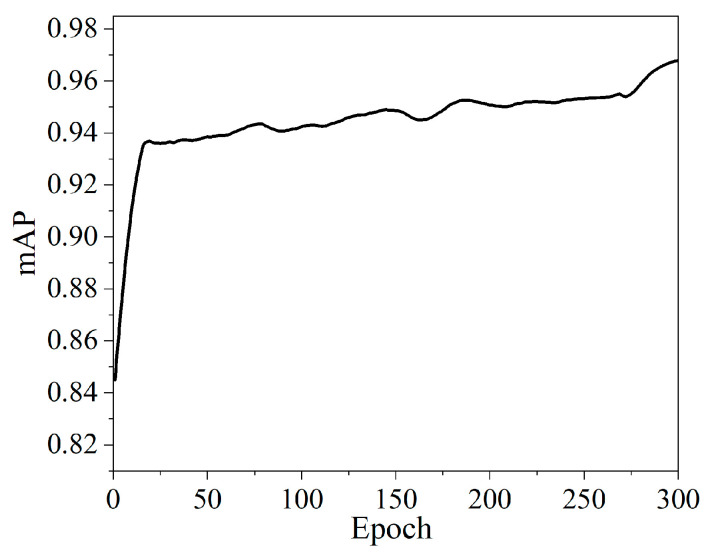
Schemes follow the same formatting.

**Figure 14 sensors-22-04364-f014:**
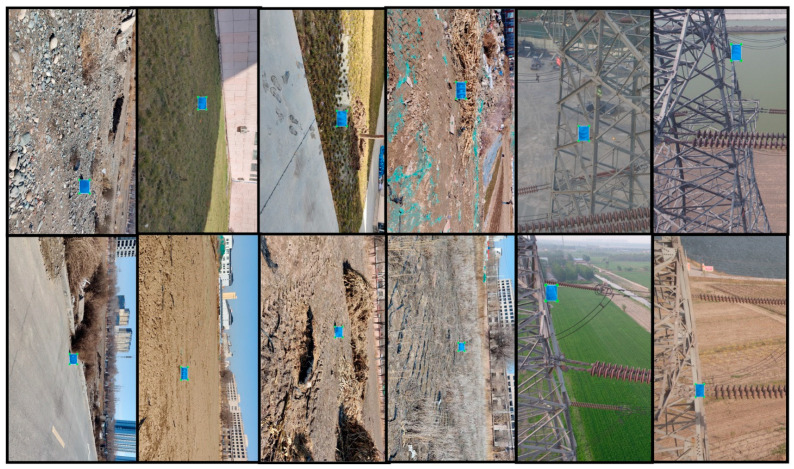
Example of the test set of images with annotations.

**Figure 15 sensors-22-04364-f015:**
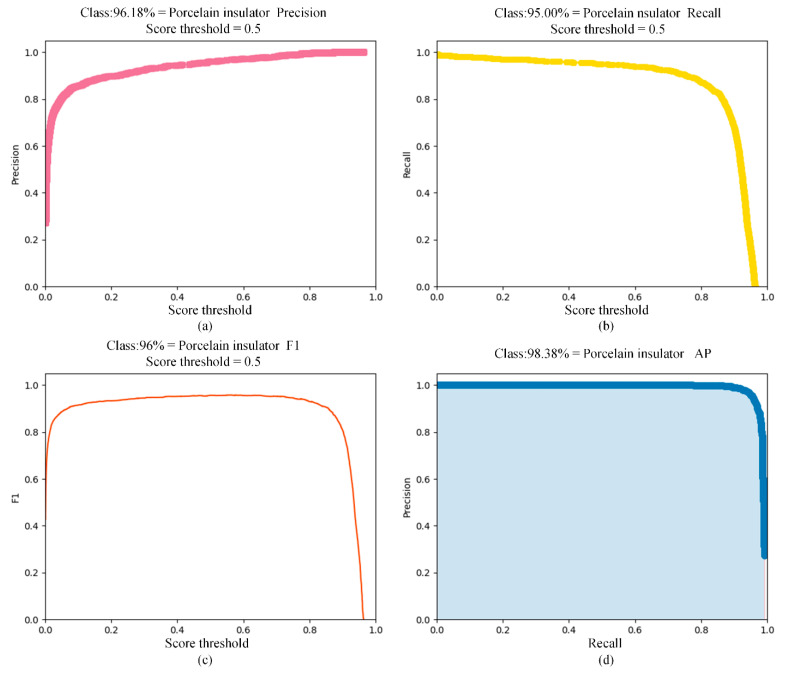
Evaluation results include P, R, F1, and AP. (**a**) P; (**b**) R; (**c**) F1; (**d**) AP.

**Table 1 sensors-22-04364-t001:** Summary of the literature for expanding dataset.

Characteristic	Core Idea	Method	Pros and Cons
Expanding a real dataset	Data augmentation [16]	Histogram equalization, gaussian blur, random translation, scaling, cutout, and rotation	Histogram equalization, random translation, and cutout have a low impact on network accuracy. Faster RCNN cannot deal with the issue of target rotation well.
	Random splicing [17]	—	The trained model used synthetic samples have a good generalization for the real images, the transparent samples made the networks difficult to converge.
	Image processing [14]	A deep Laplacian pyramid-based super-resolution network and a low-light image enhancement technique	The quality and volume of the training dataset are improved, but the images need to be collected.
Generating a synthesis dataset	Generating dataset using virtual 3D models [18,21,22]	Blender script	Dataset generation is easy and fast in a single background working condition for standard parts.
	Generating dataset based on images [19,23,24]	Mixing real and synthetic data, Generative Adversarial Networks (GANs)	The appropriate ratio is favorable for neural network learning; however, it is difficult to collect real images.
	Overlaying synthetic data on videos [20]	A ray tracing engine	Semantic segmentation datasets can be quickly generated, but the background changes are not flexible.

**Table 2 sensors-22-04364-t002:** Example of the set of prior series data.

Tower Number	Date	Time	Set of Prior Series Data
1#	2 January	10:00	*S*^1^-*P*^1^ 1/*P*^2^ 1/*P*^3^ 1/*P*^4^ 1
		15:00	*S*^1^-*P*^1^ 1/*P*^2^ 1/*P*^3^ 1/*P*^4^ 2
	3 March	13:00	*S*^1^-*P*^1^ 1/*P*^2^ 1/*P*^3^ 1/*P*^4^ 3
		18:00	*S*^1^-*P*^1^ 1/*P*^2^ 1/*P*^3^ 1/*P*^4^ 4
2#	4 June	9:00	*S*^2^-*P*^1^ 2/*P*^2^ 1/*P*^3^ 2/*P*^4^ 5
		11:00	*S*^2^-*P*^1^ 2/*P*^2^ 1/*P*^3^ 2/*P*^4^ 6

**Table 3 sensors-22-04364-t003:** The key parameters of the synthetic dataset.

Description	Symbol	Unit	Value
x_i_ axis conversion distance of insulator	*t* _x_	m	[0.3, 30]
y_i_ axis conversion distance of insulator	*t* _y_	m	{1.9, 2.2, 2.3}
z_i_ axis conversion distance of insulator	*t* _z_	m	[0, 4.6]
Translation distance x_r_ axis increase in flight	*t*′_x_	m	[−30, 30]
Translation distance y_r_ axis increase in flight	*t*′_y_	m	[−10, 10]
Translation distance z_r_ axis increase in flight	*t*′_z_	m	[−10, 10]
y_i_ axis rotation angle of insulator	*β*	°	[−90, 90]
z_i_ axis rotation angle of insulator	*γ*	°	[−60, 60]
Irradiance	*E*	W/m^2^	[0, 1032]
HDRI environmental horizontal rotation angle	*θ* _l_	°	[0, 360]
HDRI environmental vertical rotation angle	*θ* _v_	°	[−45, 45]
x_i_ axis conversion distance of insulator	*t* _x_	m	[0.3, 30]

**Table 4 sensors-22-04364-t004:** Summary of the methods for expanding dataset.

Core Idea	Method	Recognition Target	Data Volume	Sensor Type	Vehicle Type	Performance Metrics
Data augmentation [16]	Faster RCNN	Insulator	—	RGB Camera	NVIDIA RTX2080ti	Highest accuracy: 90.2%
Random splicing [17]	cGAN	Insulator	8 K	Infrared camera and RGB camera	NVIDIA GTX1080	Highest accuracy: 85%
Image processing [14]	YOLO v4	Insulator	15 K	RGB Camera	NVIDIA GTX1060	AP: 82.9%
Generating dataset using virtual 3D models [18]	CNN	Screw, nut, and washer	—	RGB Camera	Samsung S7, Epson Moverio M350	Accuracy: 91% to 99%
Mixing real and simulated samples [19]	CNN	Insulator	18 K	RGB Camera	Omnisky SCW4750 workstation	Best mixing ratio: 2.0 Accuracy: 97.9%
Overlaying synthetic wires on flight videos [20]	CNN	Wire	68 K	RGB Camera	NVIDIA Jetson TX2	AP: 73%
Generating dataset base prior series data	YOLOX	Insulator	10.8 K	D 435i	NVIDIA GTX1660	AP: 98%

## Data Availability

We published our dataset at https://github.com/zjlanthe/synthetic_data_insulator_blender, accessed on 12 April 2022.

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
