# Peer review of "A Novel Auto-Synthesis Dataset Approach for Fitting Recognition Using Prior Series Data"

_sensors, 2022, doi:10.3390/s22124364_

Round 1

Reviewer 1 Report

This paper introduces an auto-synthesis dataset method to fit recognition using prior series data. The synthetic dataset produced by this approach is validated on the YOLOX model. Overall, the paper is well written and structured, I have only some minor comments to further improve it:

1) The paper should be proofread by a native speaker to improve the quality of English and correct the typos and grammatical issues.

2) The literature review is very terse. A more comprehensive study of the sate-of-the-art should be conducted. Existing methods should  be summarized in a table along with describing used methodologies, their characteristics, pros and cons, etc.

3) The drawback and limitations of the proposed scheme need to be identified and highlighted in the Conclusion before deriving the future work.

Reviewer 2 Report

The paper is an application of combibation of known methods. The puposed method is not presented adequetly. I think the current status of the paper is not reader friendly. The authors should present complete method of the paper as an abtact method flowchart  from scratch. Also the classification problem must be visualized. Classification inputs outputs and architecture of the learning algorithm must be depicted. The literature must be feed by current related works and main scientific contributions must be emphasized by discussion of these literature. Please add an contribution table which compares the current related papers. The table, at least, must consist ml method, sensor type, vehicle type and performance metrics

Reviewer 3 Report

The developed system is very interesting and applied all over the world, it brings together different very current technologies, such as controlling the  drones, artificial intelligence and vision systems applied to data monitoring energy distribution network.

Round 2

Reviewer 2 Report

In the study, it seems that all kinds of techniques were used and tried to be published. The authors don't seem to have a chance to make it better either. The article is acceptable and printable in its current form.